# Influence of Renal Impairment and Genetic Subtypes on Warfarin Control in Japanese Patients

**DOI:** 10.3390/genes12101537

**Published:** 2021-09-28

**Authors:** Tomotaka Tanaka, Masafumi Ihara, Kazuki Fukuma, Haruko Yamamoto, Kazuo Washida, Shunsuke Kimura, Akiko Kada, Shigeki Miyata, Toshiyuki Miyata, Kazuyuki Nagatsuka

**Affiliations:** 1Department of Neurology, National Cerebral and Cardiovascular Center, Osaka 564-8565, Japan; ihara@ncvc.go.jp (M.I.); kazblues77@ncvc.go.jp (K.F.); washida@ncvc.go.jp (K.W.); kimura.shunsuke@ncvc.go.jp (S.K.); nagatuka@ncvc.go.jp (K.N.); 2Center for Advancing Clinical and Translational Sciences, National Cerebral and Cardiovascular Center, Suita, Osaka 564-8565, Japan; harukoya@ncvc.go.jp; 3Clinical Research and Development, National Cerebral and Cardiovascular Center, Suita, Osaka 564-8565, Japan; akiko.kada@nnh.go.jp; 4Department of Clinical Laboratory Medicine, National Cerebral and Cardiovascular Center, Suita, Osaka 564-8565, Japan; s-miyata@jrc.or.jp; 5Department of Cerebrovascular Medicine, National Cerebral and Cardiovascular Center, Suita, Osaka 564-8565, Japan; miyata@ncvc.go.jp

**Keywords:** warfarin, pharmacogenetics, dosing algorithm, renal impairment, *CYP2C9*, *VKORC1*

## Abstract

The genotypes of vitamin K epoxide reductase complex 1 (*VKORC1*) and cytochrome P450 2C9 (*CYP2C9*) can influence therapeutic warfarin doses. Conversely, nongenetic factors, especially renal function, are associated with warfarin maintenance doses; however, the optimal algorithm for considering genes and renal dysfunction has not been established. This single-center prospective cohort study aimed to evaluate the factors affecting warfarin maintenance doses and develop pharmacogenetics-guided algorithms, including the factors of renal impairment and others. To commence, 176 outpatients who were prescribed warfarin for thromboembolic stroke prophylaxis in the stroke center, were enrolled. Patient characteristics, blood test results, dietary vitamin K intake, and *CYP2C9* and *VKORC1 (-1639G>A)* genotypes were recorded. *CYP2C9* and *VKORC1 (-1639G>A)* genotyping revealed that 80% of the patients had *CYP2C9 *1/*1* and *VKORC1* mutant AA genotypes. Multiple linear regression analysis demonstrated that the optimal pharmacogenetics-based model comprised age, body surface area, estimated glomerular filtration rate (eGFR), genotypes, vitamin K intake, aspartate aminotransferase levels, and alcohol intake. eGFR exercised a significant impact on the maintenance doses, as an increase in eGFR of 10 mL/min/1.73 m^2^ escalated the warfarin maintenance dose by 0.6 mg. Reduced eGFR was related to lower warfarin maintenance doses, independent of *VKORC1* and *CYP2C9* genotypes in Japanese patients.

## 1. Introduction

Ever since the development of effective administration techniques for direct oral anticoagulants (DOACs), which carry a lower risk of bleeding than warfarin, warfarin has become obsolete as a mainstream anticoagulant therapy; however, it remains an important anticoagulant for patients with mechanical heart valves and those requiring hemodialysis—given the lack of clear evidence for DOAC usage. Warfarin has a narrow therapeutic range and demonstrates a wide dose variation among patients because of numerous environmental and genetic factors that influence warfarin pharmacokinetics and pharmacodynamics [1,2,3]. Therefore, determining the appropriate dose for warfarin therapy may be difficult, and physicians should perform frequent therapeutic monitoring of the effects of warfarin [4]. An insufficient warfarin dosage can cause ischemic stroke, whereas excessive dosage increases the risk of bleeding [5].

Genotyping vitamin K epoxide reductase complex 1 (*VKORC1*) and cytochrome p450 2C9 (*CYP2C9*) improves the warfarin dose adjustment. In August 2007, the US Food and Drug Administration updated the warfarin product label to include pharmacogenetic information [6]. The genes of cytochrome *P450 4F2* (*CYP4F2*) and γ glutamyl carboxylase (*GGCX*), as well as concomitant medications (i.e., amiodarone, levofloxacin, azole antifungals, rifampin, and some statins), have contributed to the dose requirements of warfarin [7,8,9]. Several studies have integrated clinical and genetic factors based on regression analysis models to predict dosage [10]. While these algorithms accurately predict warfarin dosage, Asian patients appear to benefit less from genetic-based dose calculations. The variability in warfarin dose explained by the published International Warfarin Pharmacogenetics Consortium pharmacogenetic algorithm was higher in Caucasians than in Asians (40.03% vs. 23.94%, R^2^ values) [11]. In addition, many people in the Japanese population have the same genotypes; the GG genotype of the *VKORC1 (-1639G>A)* occurs at a frequency of <1%, while the GA and AA genotypes are observed in 15% and 80% of cases, respectively. A common variant, *VKORC1-1639A,* is associated with high warfarin sensitivity and reduced dose requirements. Moreover, the frequencies of the *1/*1, *1/*3, and *3/*3 genotypes in the *CYP2C9* gene are 95%, 5%, and 0.3%, respectively [12,13,14]. The *CYP2C9*3* variant allele has been shown to decrease the enzymatic activity of *CYP2C9*, which contributes to low warfarin dose requirements. Previously, warfarin management was similar for patients with renal dysfunction and the general population [15]. However, past studies have demonstrated the influence of renal impairment on warfarin dosing [16,17,18]. Moderate and severe kidney impairments have been associated with a reduction in warfarin dose requirements. Although numerous pharmacogenetic warfarin dosing algorithms have been published [10,19], those considering renal dysfunction have not been established. The aims of this study were to identify the relationship between renal dysfunction and warfarin dose, and to develop a dosing algorithm for warfarin to predict a therapeutic dose for patients with renal dysfunction.

## 2. Materials and Methods

### 2.1. Participants and Settings

This single-center prospective cohort study was conducted from 1 March to 31 May, 2008 at a stroke center. The inclusion criteria were as follows: (1) warfarin administration for anticoagulation treatment for thromboembolic stroke prophylaxis and measurement of the prothrombin time-international normalized ratio (PT-INR) at least once every 2 months at the Department of Stroke at our institute; (2) independent living (modified Rankin Scale < 3); (3) normal intake of food through the mouth; and (4) ability to provide written informed consent. If the patient was unable to sign because of aphasia or motor dysfunction but clearly expressed consent to participate, then, a family member acted as a proxy signer. The exclusion criteria were as follows: (1) signs of chronic systemic infection; (2) severe gastrointestinal dysfunction; (3) concurrent diseases that may affect warfarin control, including malignancy or severe hepatic dysfunction; (4) suspected congenital coagulopathy; (5) thrombocytopenia < 100 × 10^9^/L; (6) history of warfarin interruption for more than 3 consecutive days in the past 2 months; and (7) mechanical cardiac valve replacement.

### 2.2. Evaluation of Demographics

The following patient characteristics were assessed: age; sex; height; body weight; body surface area (BSA), calculated using the DuBois formula [20]; current and past smoking habits; drinking habits; history of thromboembolism; and concomitant medication, including antiplatelets, statins, antibiotics, and non-steroidal anti-inflammatory drugs. Vitamin K dietary intake was assessed using a self-administered semi-quantitative food frequency questionnaire (FFQ) that contained 138 food items and 14 supplementary questions concerning the use of dietary supplements, dietary habits, and other factors [21]. The patients were requested to complete the FFQ per their memories of dietary intake in the last month. The completed questionnaires were inspected by one person. Logically incorrect answers, incomplete records, or double records were clarified from the patients and corrected. Additionally, PT-INR and warfarin daily dose before and after completing the FFQ were collected. The serum aspartate aminotransferase (AST), blood urea nitrogen (BUN), and creatinine levels were determined from peripheral whole blood samples. The estimated glomerular filtration rate (eGFR) was calculated using the Modification of Diet in Renal Disease equation with coefficients modified for Japanese patients as follows [22]: eGFR (mL/min/1.73 m^2^) = 194 × Creatinine^−1.094^ × age^−0.287^ (×0.739, if female).

### 2.3. Genetic Subtyping

Peripheral blood samples were collected after obtaining written informed consent from the participants. Among the genetic variants of the *CYP2C9* gene that are associated with warfarin dose, Asian populations do not have the *CYP2C9*2* allele but the *CYP2C9*3* allele [23]. Accordingly, genotyping of *CYP2C9*3* and *VKORC1 (-1639G>A)* was performed. The genotypes of single-nucleotide polymorphisms were identified using TaqMan PCR system (Applied Biosystems, Foster City, CA, USA).

### 2.4. Statistical Analysis

Values are expressed as means ± standard deviations or as percentages. A multiple linear regression analysis was performed to determine the effect of variables on warfarin maintenance doses. Age, BSA, eGFR, *VKORC1*, and *CYP2C9* were set as fixed variables based on univariable analysis (*p* < 0.05). Other variables potentially associated with warfarin dosing according to the univariable analysis (*p* < 0.15) were included in the multivariable analysis. Different combinations of these variables were used to evaluate the agreement between the predicted dose from the model and the observed dose using the Akaike’s information criterion (AIC), R^2^, and adjusted R^2^ value [24]. Statistical analyses were performed using SAS version 9.1 (SAS Institute Inc., Cary, NC, USA).

## 3. Results

### 3.1. Patient Characteristics

A total of 176 patients, who were prescribed warfarin for thromboembolic stroke prophylaxis, were enrolled in the study. The participant characteristics are summarized below in Table 1. Warfarin was prescribed for primary or secondary prevention of cardioembolic stroke (atrial fibrillation, 56.3%), for secondary prevention of embolic stroke caused by an undetermined source (36.4%), and for paradoxical cerebral embolism (7.3%). In total, 143 patients had a stroke history (80.8%). The mean eGFR was 65.04 mL/min/1.73 m^2^.

### 3.2. The Genotypes of CYP2C9 and VKORC1 and Their Associations with the Warfarin Maintenance Dose

One patient dropped out of the study before the genotype analysis was performed. Genotyping revealed that 80% of the patients had wild-type *CYP2C9* and *VKORC1* (Table 2). In most patients, warfarin maintenance doses were controlled by n a low-dose regimen because previous reports in our chosen hospital showed a PT-INR range of 1.6–2.6 to be safe and effective in Japanese patients with non-valvular atrial fibrillation [25,26]. Hence, in the present clinical settings, we decided to define the warfarin maintenance dose at a PT-INR range of 1.6–2.6. The distribution of the warfarin maintenance dose according to *VKORC1* and *CYP2C9* genotypes is presented in Figure 1. The mean warfarin maintenance dose in the wild-type *CYP2C9* and *VKORC1* group (*n* = 140) was 3.0 ± 0.9 mg/day. During the follow-up period, two adverse events (two cases of ischemic stroke) were recorded. No patient reported hemorrhagic adverse event.

### 3.3. Relationship between the Warfarin Maintenance Dose and the eGFR

In the univariable analysis, no significant relationship was found between vitamin K intake and warfarin maintenance doses; however, age, height, BSA, eGFR, and genetic type (VC2: *CYP2C9 *1/*1*, *VKORC1 (-1639G>A) GA* or *GG*) were found to be significantly associated with the warfarin maintenance dose (*p* < 0.01, Table 3). 

Figure 2A shows the relationship between warfarin maintenance dose and eGFR among genotypes. In particular, eGFR was significantly associated with the warfarin maintenance dose in the wild type *CYP2C9* and *VKORC1* group (*n* = 140, VC1: *CYP2C9 *1/*1, VKORC1 (-1639G>A) AA*) (Figure 2B).

### 3.4. Determining the Factors Associated with the Warfarin Maintenance Dose

The clinical variables that were potentially associated with warfarin maintenance doses were age, height, weight, BUN, creatinine, BSA, AST, white blood cell (WBC) count, vitamin K dietary intake, eGFR, and genetic types in univariable analysis (as indicated by *p*-value < 0.15, Table 3). Multiple linear regression analysis was performed to investigate the factors that determine the warfarin maintenance dose. Age, BSA, eGFR, and genotype were fixed, and the other variables were determined.

The smallest AIC was found in the model, including age, BSA, eGFR, genotype, vitamin K intake, AST, and alcohol intake (AIC = 411.9, R^2^ = 0.561, adjusted R^2^ = 0.533, Table 4). The largest adjusted R^2^ was found in the model, including age, BSA, eGFR, genotype, vitamin K intake, AST, alcohol intake, and WBC count (AIC = 412.9, R^2^ = 0.589, adjusted R^2^ = 0.558). Although the WBC count was the only different variable between the two models, its effect on warfarin maintenance doses was very small because an increase in the WBC count of 100 WBCs/µL would decrease the warfarin maintenance dose by 0.01 mg. Therefore, we preferred the first model. The eGFR exerted a large effect size because an increase of 10 mL/min/1.73 m^2^ could lead to an increase in the warfarin maintenance dose by 0.6 mg.

### 3.5. Pharmacogenetics-Guided Warfarin Dosing Algorithm for Predicting the Therapeutic Dose

The following algorithm was devised to predict the warfarin maintenance dose:Dose (mg/day) = 1.096 − 0.217 × age (/10 years) + 0.015 × BSA (/100 m^2^) + 0.201 × eGFR (/10 mL/min/1.73 m^2^) + 1.886 × VC2 ^†^ − 1.164 × VC3 ^‡^ + 0.601 × VC4 ^¶^ + 0.011 × VK (/10 µg/day) − 0.021 × AST (IU/L) + 0.186 × drinking
VC2 ^†^: *CYP2C9 *1/*1, VKORC1 (-1639G>A) GA* or *GG*VC3 ^‡^: *CYP2C9 *1/*3, VKORC1 (-1639G>A) AA*VC4 ^¶^: *CYP2C9 *1/*3, VKORC1 (-1639G>A) GA* or *GG*

## 4. Discussion

In this study, we found that the eGFR independently influenced the maintenance dose of warfarin. As renal function deteriorated, the warfarin dose required to maintain the INR within the therapeutic range decreased. As age and BSA were associated with increases in the individual variable response to treatment [27,28], this relationship was independent of the other clinical and genetic factors. 

Previous studies have demonstrated the influence of renal impairment on the warfarin maintenance dose [16,17,18]. Moreover, the influence of reduced kidney function was associated with lower warfarin doses after accounting for clinical and genetic factors [29]. Compared with no/mild renal dysfunction (eGFR, ≥60 mL/min/1.73 m^2^), patients with moderate renal dysfunction (eGFR, 30–59 mL/min/1.73 m^2^) required 9.5% lower doses, while those with severe renal dysfunction (eGFR, <30 mL/min/1.73 m^2^) required 19.1% lower doses. These data suggested that warfarin dosing based on standard genotypic and clinical information may be insufficient for optimal dosing in patients with renal dysfunction. Our findings agree with these data.

The exact mechanism by which renal injury results in cytochrome P450 inactivation has not been elucidated. Several studies have revealed that patients with renal dysfunction exhibit reduced non-renal clearance and altered bioavailability of drugs predominantly metabolized by the liver, perhaps reflecting a selective decrease in hepatic p450 activity [15,30,31,32]. Another study demonstrated that parathyroid hormone was associated with p450 downregulation in chronic renal failure cases [33]. Furthermore, one study reported the inactivation of the p450 super-family in acute kidney injury [34]. Further research is required to completely understand the mechanism of p450 inactivation in renal dysfunction. Moreover, the prevalence of atrial fibrillation in patients requiring long-term hemodialysis is extremely high, reaching 27%, and it is associated with increased mortality [35]. Additionally, patients with severe renal dysfunction requiring hemodialysis who receive warfarin exhibit an increased risk of bleeding. In such patients, not only is the bleeding time prolonged and the factors responsible for coagulation, anticoagulation, and fibrinolysis altered, but the risk of bleeding is higher [36]. Limdi et al. also indicated that reduced renal function was associated with a higher incidence of over-anticoagulation (PT-INR > 4) [16]. Therefore, warfarin use without adequate monitoring is dangerous, particularly in patients with severe renal dysfunction.

Our study suggested that the contribution of dietary vitamin K intake to the warfarin dose is apparently small. The plausible reason is that vitamin K is synthesized by the normal flora of the human intestine; therefore, there can be some discrepancy between the actual vitamin K content in the human body and estimated vitamin K levels measured by food intake [37]. In addition, the most accurate algorithm included AST. The link between AST and warfarin maintenance doses was not reported in previous studies, but AST was believed to be associated with liver function for metabolizing warfarin. The factors that reflect liver function with relation to warfarin metabolism must be examined more precisely.

This study had several limitations. First, as the variant *CYP2C9* and wild-type *VKORC1* are relatively rare in the Japanese population, we combined some genotype groups (VC2, 3, 4) for multiple linear regression analysis, despite only one genetic group (VC2: *CYP2C9 *1/*1, VKORC1 (-1639G>A) GA* or *GG*) being significantly associated with the warfarin maintenance dose. Moreover, we did not estimate the sample size *a priori*; in particular, it was difficult to perform a power analysis for validity. Second, the PT-INR target range in our hospital was lower than the range recommended by international guidelines (PT-INR, 2.0–3.0) [38]. We could not modify the PT-INR target range for this study because it was an observational study. The value of β for each independent variable in the multiple linear regression analysis would be higher if the PT-INR target range was high. Third, vitamin K dietary intake was assessed using the FFQ, which is useful in estimating the vitamin K content of food; however, the absorption of vitamin K is influenced by gastrointestinal conditions. Gastrointestinal illnesses, such as vomiting or diarrhea, affect the absorption of vitamin K [39]. Therefore, we should determine the actual concentration of serum vitamin K. We were also unable to check the *CYP4F2* variants—which is another common genetic covariate in Asians [40]—nor were we able to validate our pharmacogenetics-guided warfarin-dosing algorithm. However, we are conducting another multicenter study on warfarin use to reveal the efficacy of the algorithm. Finally, we did not assess renal function using other biomarkers, such as the cystatin level, which have been demonstrated to predict kidney function with greater accuracy [41]. Plasma creatinine is related to muscle mass and is affected by age and sex. As plasma concentrations increase, the tubular secretion of creatinine increases, leading to an overestimation of GFR. Therefore, we can obtain a more accurate and precise relationship between renal function and the warfarin maintenance dose using cystatin levels.

## 5. Conclusions

In addition to the conventional factors, reduced eGFR was related to lower warfarin maintenance doses. We can estimate the accurate requirements of warfarin maintenance doses by considering BSA, age, change in vitamin K intake, AST level, drinking habits, genotype, and eGFR in Japanese patients. Further exploration is necessary to confirm the association between these factors and warfarin doses more precisely.

## Figures and Tables

**Figure 1 genes-12-01537-f001:**
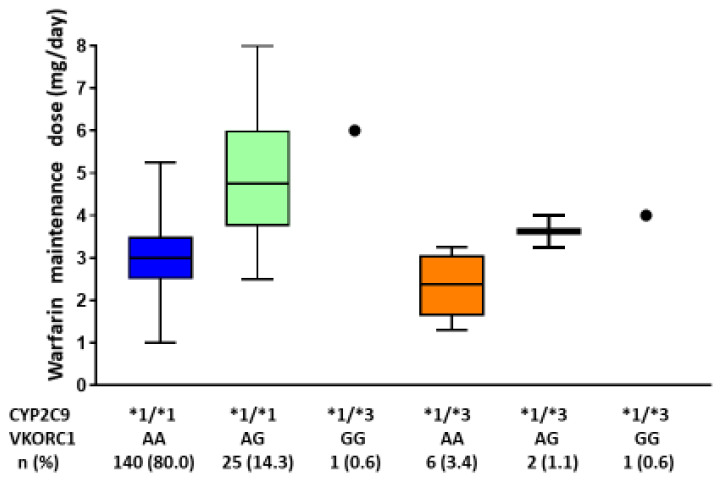
Warfarin maintenance dose stratified by *CYP2C9* and *VKORC1* genotypes. Warfarin maintenance dose was determined as the dose required to maintain PT-INR values within the target therapeutic range (1.6–2.6) during the follow-up period. In box-and-whisker plots, the central horizontal bars indicate the median values, and the lower and upper boundaries show the 25th and 75th percentiles, respectively. PT-INR, the prothrombin time-international normalized ratio.

**Figure 2 genes-12-01537-f002:**
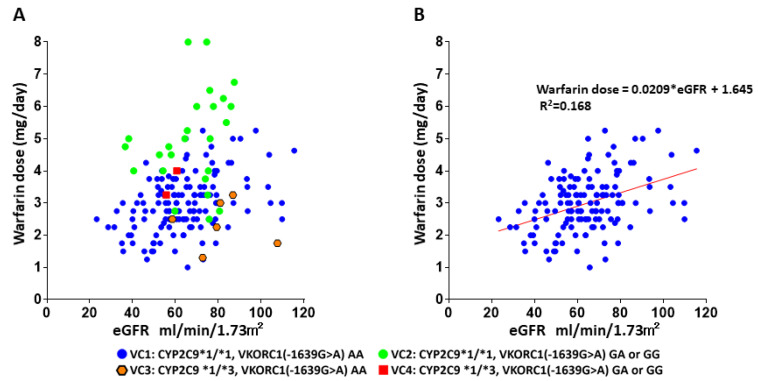
Scatter plots show the relationship between warfarin maintenance dose and eGFR: (**A**) Different symbols are used to distinguish among genotypes (blue plots, VC1: *CYP2C9 *1/*1* and *VKORC1 [-1639G>A] AA*; light green plots, VC2: *CYP2C9 *1/*1* and *VKORC1 [-1639G>A] GA* or *GG*; orange plots, VC3: *CYP2C9 *1/*3* and *VKORC1 [-1639G>A] AA*; red plots, VC4: *CYP2C9 *1/*3* and *VKORC1 [-1639G>A] GA* or *GG*). (**B**) The red line shows a linear regression line for the only VC1 group (*n* = 140, *CYP2C9 *1/*1* and *VKORC1 [-1639G>A] AA*). The warfarin maintenance dose = 0.0209*eGFR [mL/min/1.73 m^2^] + 1.645, R^2^ = 0.168, *p* value < 0.001). eGFR: estimated glomerular filtration rate.

**Table 1 genes-12-01537-t001:** Patient characteristics.

Variables	*N* = 176
Age (years, mean ± SD)	72.1 ± 8.1
Male (*n*, %)	126 (71.6%)
Body height (cm, mean ± SD)	162.4 ± 8.2
Body weight (kg, mean ± SD)	61.2 ± 10.5
BSA (m^2^, mean ± SD)	1.65 ± 0.17
Current smokers (*n*, %)	8 (4.5%)
Current alcohol consumption (*n*, %)	86 (48.9%)
History of embolic event (*n*, %)	17 (9.7%)
Concomitant medication	
Aspirin use (*n*, %)	30 (17.0%)
Another antiplatelet drugs use (*n*, %)	5 (2.8%)
Statin use (*n*, %)	62 (35.2%)
PT-INR (mean ± SD)	2.0 ± 0.4
WBC (count/L, mean ± SD)	5.9 ± 1.6 × 10^9^
Hemoglobin (g/dL, mean ± SD)	13.7 ± 1.5
AST (IU/L, mean ± SD)	26.3 ± 8.2
ALT (IU/L, mean ± SD)	20.7 ± 11.5
BUN (mg/dL, mean ± SD)	17.1 ± 5.1
Creatinine (mg/dL, mean ± SD)	0.9 ± 0.3
eGFR (mL/min/1.73 m^2,^ mean ± SD)	65.04 ± 16.98

SD, standard deviation; BSA, body surface area; PT-INR, prothrombin time-international normalized ratio; WBC, white blood cell; AST, aspartate aminotransferase; ALT, alanine aminotransferase; BUN, blood urea nitrogen; eGFR, estimated glomerular filtration rate; Data are presented as mean (SD) or absolute (percentage) values.

**Table 2 genes-12-01537-t002:** *CYP2C9* and *VKORC1* genotypes.

	*VKORC1* Genotype (-1639G>A)		
*CYP2C9* Genotype	AA	GA	GG
*1/*1	140 (80.0%)	25 (14.3%) ^†^	1 (0.6%) ^†^
*1/*3	6 (3.4%) ^‡^	2 (1.1%) ^¶^	1 (0.6%) ^¶^
*3/*3	0	0	0

The genetic analysis could not be performed in one patient. Genotype frequencies for the *CYP2C9* and *VKORC1* genes are divided into four groups. VC1 (reference group): *CYP2C9 *1/*1*, *VKORC1 (-1639G>A) AA* (*n* = 140). VC2 ^†^: *CYP2C9 *1/*1, VKORC1 (-1639G>A) GA* or *GG* (*n* = 26). VC3 ^‡^: *CYP2C9 *1/*3, VKORC1 (-1639G>A) AA* (*n* = 6). VC4 ^¶^: *CYP2C9 *1/*3, VKORC1 (-1639G>A) GA* or *GG* (*n* = 3). No subject carried the *CYP2C9 *3/*3* in this study.

**Table 3 genes-12-01537-t003:** Univariable associations between variables and the warfarin maintenance dose.

Variables	β Value	Standard Error	*p* Value
Sex	−0.196	0.208	0.348
Age	−0.043	0.011	0.0001 **
Height	0.041	0.011	0.0004 **
Weight	0.022	0.009	0.016 *
BSA	0.00017	0.00005	0.0023 **
BUN	−0.041	0.018	0.022 *
Creatinine	−0.806	0.324	0.014 *
AST	−0.02	0.012	0.087
WBC	−0.0001	0.00006	0.076
Vitamin K intake	0.018	0.012	0.134
Smoking	0.252	0.439	0.567
Alcohol intake	0.209	0.188	0.268
Aspirin use	−0.138	0.247	0.578
Statin use	0.133	0.197	0.501
eGFR	0.02	0.005	0.0003 **
Genotypes			
VC2 ^†^	1.896	0.213	<0.0001 **
VC3 ^‡^	−0.984	0.499	0.050
VC4 ^¶^	0.34	0.707	0.632

BSA, body surface area; BUN, blood urea nitrogen; AST, aspartate aminotransferase; WBC, white blood cell; eGFR, estimated glomerular filtration rate. VC1 (reference group): *CYP2C9 *1/*1, VKORC1 (-1639G>A) AA*. VC2 ^†^: *CYP2C9 *1/*1, VKORC1 (-1639G>A) GA* or *GG*. VC3 ^‡^: *CYP2C9 *1/*3, VKORC1 (-1639G>A) AA*. VC4 ^¶^: *CYP2C9 *1/*3, VKORC1 (-1639G>A) GA* or *GG*. * *p* value < 0.05, ** *p* value < 0.01.

**Table 4 genes-12-01537-t004:** Factors affecting the warfarin maintenance dose in multiple linear regression model.

Variables	β Value	Standard Error	*p* Value
age (/10 years)	−0.217	0.103	0.037
BSA (/100 m^2^)	0.015	0.005	0.002
eGFR (/10 mL/min/1.73 m^2^)	0.201	0.047	<0.001
Genotypes			
VC2 ^†^	1.886	0.189	<0.001
VC3 ^‡^	−1.164	0.406	0.005
VC4 ^¶^	0.601	0.633	0.344
Vitamin K intake (µg/day)	0.011	0.009	0.225
AST (IU/L)	−0.021	0.009	0.029
Alcohol intake	0.186	0.148	0.210

AIC = 411.9, R^2^ = 0.561, Adjusted R^2^ = 0.533. BSA, body surface area; AST, aspartate aminotransferase; eGFR, estimated glomerular filtration rate. VC2 ^†^: *CYP2C9 *1/*1, VKORC1 (-1639G>A) GA* or *GG*. VC3 ^‡^: *CYP2C9 *1/*3, VKORC1 (-1639G>A) AA*. VC4 ^¶^: *CYP2C9 *1/*3, VKORC1 (-1639G>A) GA* or *GG*.

## Data Availability

Data will be provided upon reasonable request from the corresponding author.

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
