# Peer review of "Influence of Renal Impairment and Genetic Subtypes on Warfarin Control in Japanese Patients"

_genes, 2021, doi:10.3390/genes12101537_

Round 1

Reviewer 1 Report

In essence the authors have combined genetic and biochemical data on kidney function to develop an algorithm to predict dosage of warfarine. The reviewer is curious to know wether this algorithm would work to improve dosing of warfarine at least as demonstrated in a few examples.

Author Response

Responses to Reviewer 1

In essence the authors have combined genetic and biochemical data on kidney function to develop an algorithm to predict dosage of warfarine. The reviewer is curious to know wether this algorithm would work to improve dosing of warfarine at least as demonstrated in a few examples.

We would like to thank the reviewer for evaluating our manuscript and for his/her insightful suggestion. However, we did not try to estimate the warfarin maintenance dose using our algorithm. To reveal the efficacy of the algorithm, we are conducting a multicenter study. However, it is difficult to present the findings. We have discussed this issue as a limitation in the revised manuscript as follows:

Lines 287–290

“Fifth, we did not validate our pharmacogenetics-guided warfarin-dosing algorithm. However, we are conducting another multicenter study on warfarin use to reveal the efficacy of the algorithm.”

Reviewer 2 Report

Tanaka et al. in this work evaluated in Japanise patients the relationship between renal dysfunction (eGFR  reduction) and warfarin dosage independently from pharmacogenomic evaluation developing a predictive dosing algorithm for warfarin therapeutic dosage in patients with renal dysfunction. The study is well designed and authors put in evidence its critical points. However, figures 1 and 2 must be modified and made more interesting and easier to understand, especially figure 2. Also the legend of the figures must be clearer.

Author Response

Responses to Reviewer 2

Tanaka et al. in this work evaluated in Japanise patients the relationship between renal dysfunction (eGFR  reduction) and warfarin dosage independently from pharmacogenomic evaluation developing a predictive dosing algorithm for warfarin therapeutic dosage in patients with renal dysfunction. The study is well designed and authors put in evidence its critical points. However, figures 1 and 2 must be modified and made more interesting and easier to understand, especially figure 2. Also the legend of the figures must be clearer.

We would like to thank the reviewer for evaluating our manuscript and for his/her insightful suggestion. Please note that we have recreated Figures 1 and 2, and added more explanations to make it easier to understand.

Reviewer 3 Report

The authors have reported the effects of renal function and pharmacogenetics (2C9/VKORC1) on warfarin dose control.

Abstract:

Lines 17-19: This is incorrect. Several studies have been done. This should be corrected. See (https://pubmed.ncbi.nlm.nih.gov/33080066/).

Introduction

Line 43: “frequent blood analysis” should be replaced with therapeutic drug monitoring and elsewhere in the manuscript.

Lines 46- 48. This is a good place to discuss other genetic factors, such as GGCX and CYP4F2. Other factors to consider are concomitant medications such as amiodarone, statins, azole drugs. This should be discussed on how it can impact warfarin dosing.

Lines 53: “Whites” this is not an appropriate word to use. Rather define it by Country or region. So European or , American, or African American. Use the ethnic definitions as used by the 1000genomes consortium.

Methods & Results – good, no comments.

Discussion: Authors should do a power analysis and comment on their small cohort.

Author Response

Responses to Reviewer 3

The authors have reported the effects of renal function and pharmacogenetics (2C9/VKORC1) on warfarin dose control.

Abstract:

Lines 17-19: This is incorrect. Several studies have been done. This should be corrected. See (https://pubmed.ncbi.nlm.nih.gov/33080066/).

The authors would like to thank the reviewer for his/her constructive critique to improve the manuscript. We have made every effort to address the issues raised and to respond to all comments. The revisions are indicated highlighted in the revised manuscript. Please note that we have revised this sentence as follows:

Lines 17-19

“Conversely, nongenetic factors, especially renal function, are associated with warfarin maintenance doses; however, the optimal algorithm considering the genes and renal dysfunction has not been established.”

 Introduction

Line 43: “frequent blood analysis” should be replaced with therapeutic drug monitoring and elsewhere in the manuscript.

We would like to thank the reviewer for the comment. Please note that we have revised this sentence as follows:

Lines 42-43

“Therefore, determining the appropriate dose for warfarin therapy could be difficult, and physicians should perform frequent therapeutic monitoring of warfarin effect [4].”

 Lines 46- 48. This is a good place to discuss other genetic factors, such as GGCX and CYP4F2. Other factors to consider are concomitant medications such as amiodarone, statins, azole drugs. This should be discussed on how it can impact warfarin dosing.

We would like to thank the reviewer for the insightful suggestion. Please note that we have provided the explanations as follows:

Lines 49-52

“Additionally, the genes of cytochrome P450 4F2 (CYP4F2) and gamma glutamyl carboxylase (GGCX), as well as concomitant medications (i.e., amiodarone, levofloxacin, azole antifungals, rifampin, and some statins) also contributed to the dose requirements of warfarin [7-9].”

 Lines 53: “Whites” this is not an appropriate word to use. Rather define it by Country or region. So European or , American, or African American. Use the ethnic definitions as used by the 1000genomes consortium.

We would like to thank the reviewer for the comment. Please note that we have revised this word in the manuscript.

Methods & Results – good, no comments.

Discussion: Authors should do a power analysis and comment on their small cohort.

We asked our statistician concerning power analysis. However, as our study was an observational pilot study, we did not estimate sample size a priori. Thus, it is difficult to perform a power analysis. We have discussed this issue as a limitation in the Discussion section as follows:

Lines 273-274

“Moreover, we did not estimate the sample size a priori; especially, it was difficult to perform a power analysis for the validity.”